# Exploring Cationic Guar Gum: Innovative Hydrogels and Films for Enhanced Wound Healing

**DOI:** 10.3390/pharmaceutics16091233

**Published:** 2024-09-22

**Authors:** Kamila Gabrieli Dallabrida, Willer Cezar Braz, Crisleine Marchiori, Thainá Mayer Alves, Luiza Stolz Cruz, Giovanna Araujo de Morais Trindade, Patrícia Machado, Lucas Saldanha da Rosa, Najeh Maissar Khalil, Fabiane Gomes de Moraes Rego, André Ricardo Fajardo, Luana Mota Ferreira, Marcel Henrique Marcondes Sari, Jéssica Brandão Reolon

**Affiliations:** 1Departamento de Farmácia, Universidade Estadual do Centro-Oeste, Guarapuava 85040-167, PR, Brazil; kadallabrida@gmail.com (K.G.D.); contatowiller@gmail.com (W.C.B.); crisleinemarchiori@gmail.com (C.M.); mayerthainaalves@gmail.com (T.M.A.); luizastolz@gmail.com (L.S.C.); 2Centro de Estudos em Biofarmácia, Departamento de Farmácia, Programa de Pós-Graduação em Ciências Farmacêuticas, Universidade Federal do Paraná, Curitiba 80210-170, PR, Brazil; giovannaaraujo@ufpr.br (G.A.d.M.T.); patricia.machado.ufpr@gmail.com (P.M.); luanamota@ufpr.br (L.M.F.); 3Laboratório de Biomateriais, Centro de Ciências da Saúde, Departamento de Odontologia Restauradora, Universidade Federal de Santa Maria, Santa Maria 97015-372, RS, Brazil; lucas.saldanha@acad.ufsm.br; 4Applied Nanostructured Systems Laboratory, Universidade Estadual do Centro-Oeste, Guarapuava 85040-167, PR, Brazil; najehunicentro@hotmail.com; 5Grupo de Pesquisa em Doenças Metabólicas (GPDM), Departamento de Análises Clínicas, Programa de Pós-Graduação em Ciências Farmacêuticas, Universidade Federal do Paraná, Curitiba 80210-170, PR, Brazil; rego@ufpr.br; 6Laboratório de Tecnologia e Desenvolvimento de Compósitos e Materiais Poliméricos (LaCoPol), Universidade Federal de Pelotas, Campus Capão do Leão, Pelotas 96010-900, RS, Brazil; andre.fajardo@ufpel.edu.br

**Keywords:** biopolymer, biomaterials, wound dressing, solid dosage forms, natural gum

## Abstract

**Background/Objectives:** This study developed and characterized hydrogels (HG-CGG) and films (F-CGG) based on cationic guar gum (CGG) for application in wound healing. **Methods:** HG-CGG (2% *w*/*v*) was prepared by gum thickening and evaluated for pH, stability, spreadability, and viscosity. F-CGG was obtained using an aqueous dispersion of CGG (6% *w*/*v*) and the solvent casting method. F-CGG was characterized for thickness, weight uniformity, morphology, mechanical properties, hydrophilicity, and swelling potential. Both formulations were evaluated for bioadhesive potential on intact and injured porcine skin, as well as antioxidant activity. F-CGG was further studied for biocompatibility using hemolysis and cell viability assays (L929 fibroblasts), and its wound-healing potential by the scratch assay. **Results:** HG-CGG showed adequate viscosity and spreadability profiles for wound coverage, but its bioadhesive strength was reduced on injured skin. In contrast, F-CGG maintained consistent bioadhesive strength regardless of skin condition (6554.14 ± 540.57 dyne/cm^2^ on injured skin), presenting appropriate mechanical properties (flexible, transparent, thin, and resistant) and a high swelling capacity (2032 ± 211% after 6 h). F-CGG demonstrated superior antioxidant potential compared to HG-CGG (20.50 mg/mL ABTS+ radical scavenging activity), in addition to exhibiting low hemolytic potential and no cytotoxicity to fibroblasts. F-CGG promoted the proliferation of L929 cells in vitro, supporting wound healing. **Conclusions:** Therefore, CGG proved to be a promising material for developing formulations with properties suitable for cutaneous use. F-CGG combines bioadhesion, antioxidant activity, biocompatibility, cell proliferation, and potential wound healing, making it promising for advanced wound treatment.

## 1. Introduction

The skin is the largest organ in the human body and the first line of defense against harmful agents, such as mechanical trauma, UV light, and pathogens [1,2]. Changes in skin integrity resulting from trauma lead to the formation of wounds, which may be caused by external factors such as surgery, physical impact, heat, cold, chemicals, and biological factors, including blood supply disorders, diabetes, and leishmaniasis [3,4].

Carefully selecting an appropriate wound dressing is crucial to accelerate the healing process. The dressing must be non-toxic, non-sensitizing, and possess suitable mechanical and rheological properties for easy application to the affected area. Moreover, maintaining wound moisture and preventing infection are essential for promoting effective healing [4,5]. Various dressings are available for managing acute and chronic wounds, with particular attention given to hydrogels and films based on natural polysaccharides due to their potential advantages [6].

Hydrogels (HGs) are three-dimensional hydrophilic structures with a wide range of applications in the biomedical field, such as tissue engineering scaffolds, drug release platforms, and wound dressings. These semisolid pharmaceutical forms have specific characteristics that promote these applications, such as a porous structure, flexibility in fabrication, viscoelasticity, adjustable stiffness, ability to mimic natural tissue physical properties and accelerate tissue regeneration, biocompatibility, and wound protection by adhesion, and ability to store therapeutic molecules within their networks. Overall, HGs can be prepared from different polymers, but those from natural sources are the most commonly used [7,8,9,10].

Remarkably, polymeric films are innovative in treating wounds since polymeric materials have a high tensile strength, stability, and exceptional skin adhesion capabilities. These solid pharmaceutical forms consist of a thin polymeric layer, which may or may not include plasticizers, with easy application, flexibility, and a pleasant sensation. Furthermore, they offer advantages such as transparency, allowing for wound visualization, effective absorption of exudates, and the creation of a protective barrier against microbial contamination. Moreover, they provide a greater dosage precision, reduced application frequency, and superior physical–chemical stability of certain active ingredients compared to semisolid formulations [11,12,13].

Polymers, synthetic or natural, such as natural gums, are essential in the composition of HG and films. Natural gums are biocompatible, biodegradable, and non-toxic polymers with antioxidant and healing properties. Moreover, natural gums have multifunctions and the ability to gel quickly, retain water, and possess exceptional bioadhesive properties, which are fundamental for pharmaceutical applications, making them excellent candidates for HG preparation [6]. Additionally, these polymers are low-cost, easy to access, and contain reactive functional groups such as hydroxyl and amino groups that allow modifications or reactions with other substances [7,10,14].

Regarding natural gum, guar gum is extracted from the seeds of *Cyamopsis tetragonolobus* (L.) Taub. And is highlighted for its high-molecular-weight structure and ability to form viscous colloidal dispersions that resist pH variations and thermal processes. The composition of this gum includes mannose units with alternating lateral branched galactose groups. Additionally, this gum has been approved by the Food and Drug Administration (FDA, Silver Spring, MD, USA) and is applied in several formulations such as polymeric films, hydrogels, and capsules [10,15].

Chemical modifications of polysaccharides can introduce additional functionalities. Cationic guar gum (CGG) is a modified form of guar gum where hydroxyl groups are replaced with hydroxypropyltrimonium groups [16] (Figure 1). A quaternary ammonium functional group gives a positive charge to the polymer, improving its properties as a thickener and enhancer of viscosity and volume. Furthermore, a positive charge-related property may represent a potential interaction with the cell surface of possible pathogens present in infected wounds, enhancing the destruction of the integrity of the microorganisms’ cell membranes. Additionally, CGG has a superior solubility and thermal stability compared to guar gum, and the use of cationic polymers for the development of pharmaceutical preparations for cutaneous use may be interesting to increase the formulations’ bioadhesion to the skin, increasing the duration of stay at the site of application [7,16,17].

Of particular importance, the literature does not report the use of CGG alone for wound-healing HGs or films so far. Therefore, this study aimed to develop and characterize a cationic guar gum HG and polymeric film, cutaneous-friendly platforms intended to be potentially applied as wound-healing devices. Additionally, the mechanical properties of each platform, bioadhesive strength, swelling index, antioxidant activity, cytotoxicity, and in vitro wound-healing properties were assessed. This research addresses the need for new wound treatment agents and the lack of information on using CGG for formulating skin applications.

## 2. Materials and Methods

### 2.1. Materials and Reagents

Cationic guar gum (CGG), chemically known as guar hydroxypropyltrimonium chloride (CAS number: 65497-29-2), was obtained from Engenharia das Essências (São Paulo, SP, Brazil), and glycerol was supplied by Dinâmica (Indaiatuba, SP, Brazil). Imidazolidinyl urea, 2,2′-azinobis-(3-ethylbenzothiazoline-6-sulfonic acid) (ABTS), DMEM, fetal bovine serum, antibiotic, and trypsin were obtained from Gibco, and MTT (3-(4, 5-dimethylthiazol-2-yl)-2, 5-diphenyltetrazolium bromide) was acquired from Sigma-Aldrich (San Luis, MO, USA). L-929 cells (murine fibroblast) were acquired from the commercial Rio de Janeiro Cell Bank (code 0188, ATCC code CCL-1) (Duque de Caxias, RJ, Brazil).

### 2.2. Cationic Guar Gum-Based Formulations

In a preformulation step, a range of gum concentrations (1%, 2%, 3%, and 6%, *w*/*w*) was examined to select the optimal concentration for HG preparation. This evaluation considered pertinent research involving modified CGG hydrogels or their conjunction with other materials [7,9,18]. The gelation process began with thoroughly mixing CGG with deionized water until the gum was fully dispersed. Subsequently, after 30 min of thorough blending using a mortar and pestle, the process was completed. Following this, imidazolidinyl urea was incorporated as a preservative (0.2% *w*/*w*). The resulting HG was named HG-CGG and hermetically sealed and stored at room temperature for subsequent analysis.

Similarly, the composition of the film formulation was also screened by exploring different concentrations of CGG (1, 2, 4, and 6% *w*/*v*) and glycerol (2, 4, and 5% *w*/*v*). The solid form was prepared using the solvent-casting method [11]. First, glycerol was mixed with 25 mL of distilled water in a heating bath (HJ-3(ATRA) model, Ionlab, Araucária, Brazil) at 40 °C while stirring magnetically. Once the glycerol was completely dispersed, CGG was added to the solution, and the mixture was homogenized under magnetic stirring for 1 min. Both gum and glycerol concentrations were based on pilot studies, identifying the ideal proportion to achieve the desired strength and flexibility in the films. Finally, the dispersion was transferred to Petri dishes (100 mm diameter) and left to dry in an oven (ECB-150D model; Biancodent equipamentos, Araucária, Brazil) at 40 °C for 24 h. This process allowed the solvent to evaporate, resulting in the formation of the film. The films produced by CGG were referred to as F-CGG.

### 2.3. Semisolid Characterization

The HGs were characterized based on pH values, physical stability, spreadability, and viscosity. To determine pH, 1.0 g of HG was dispersed in 10.0 mL of deionized water and stirred magnetically until complete dispersion before pH determination, performed with a calibrated potentiometer (PG-1800 model, Gehaka, São Paulo, Brazil). The physical stability was determined using the centrifugation test (3500 rpm, 30 min; Centribio 80-2B model; Biosystem, Campinas, Brazil) at room temperature (25 °C) [19]. This test replicates long-term storage conditions in a shorter time frame by applying an accelerated gravitational force, allowing for the assessment of potential component separation or sedimentation. The absence of visible phase separation after centrifugation indicates that the formulations have adequate physical stability for consistent performance during storage and handling. The formulations were visually inspected for instability phenomena, such as phase separation.

Spreadability was evaluated according to Rigo and co-workers [20] based on the parallel plate method, with some modifications. An amount of HG was placed in a central hole (1 cm diameter) of glass support containing graph paper. The sample was subsequently pressed with known weights of glass plates during a one-minute interval to each added plate. After each plate was added, the average radius and diameter of the scattered area’s circumference were determined. The spreadability factor was obtained using Equation (1):(1)$$Sf\;({\mathrm{mm}}^{2}/g)=\frac{A}{W}$$
where *Sf* is the spreadability factor (mm^2^/g), *A* is the maximum spread area (mm^2^) after all plates were added, and *W* is the total weight added (g).

Finally, viscosity was determined using a Brookfield rotational viscometer (LVDV-II+ model; Brookfield Ametec, Middleborough, MA, USA) with an RS04 spindle. Approximately 30 g of HG was submitted to shear rates of 0.033, 0.118, 0.388, and 0.773 s^−1^ at room temperature (25 ± 1 °C), maintaining the ideal torque between 10 and 90%. All tests were performed in triplicate of three different batches.

### 2.4. Films’ Characterization

The films were characterized by thickness, transparency, weight homogeneity, morphology, folding endurance, mechanical properties, swelling index, and hydrophobicity. The film’s thickness was assessed with a digital micrometer by measuring five different locations on each film (four in the corners and one in the middle). Regarding transparency, the films were analyzed with a spectrophotometer (UV-1800 model, Shimadzu, Kyoto, Japan) by cutting into appropriate fragments (0.5 cm × 4.0 cm) and placed in quartz cuvettes. After that, a spectrum scan (220–800 nm) was performed. To determine the weight homogeneity, three fragments (1.0 cm^2^) from different film locations were cut and individually weighed on an analytical balance (AUW220D model; Marte Científica, Valinhos, Brazil) [21].

The films’ morphology and microstructure were assessed using a scanning electron microscope (MEV; VEGA3 model, TESCAN, Brno, Czech Republic). A cryo-fracture was obtained after the film was immersed in liquid nitrogen to visualize the lateral sections after a fracture. The samples were covered with gold and analyzed using an accelerating voltage of 15 kV.

Folding endurance was determined by repeatedly folding the film at an angle of 180° from the plane in the same place 300 times, observing the formation of grooves or the occurrence of breakage [11]. The universal testing machine (DL-2000 model, Emic, São José dos Pinhas, Brazil) was used to measure the mechanical properties of the films, including tensile strength, elongation, and Young’s modulus, by ASTM-D882-02 standards [22]. Film samples (60 mm × 45 mm) were individually fixed to the machine’s probe for testing. A tensile load (50 mm/min) was applied to estimate the film’s maximum deformation by calculating the percentage change in the length of the sample relative to its original size. Tensile strength was calculated by dividing the rupture force by the cross-sectional area of the sample and expressed in MPa. Young’s modulus was obtained by evaluating the relationship between stress and strain values. The results were expressed as the percentage for elongation. For the swelling index determination [23], dry films were cut into 1.0 cm^2^ pieces, placed in 2 mL Eppendorf tubes, and weighed. Next, 2 mL of phosphate buffer pH 5.5 was added, and the film fragment was submerged for 6, 12, and 24 h. The pH of 5.5 was chosen to mimic the skin’s natural pH, which typically ranges from 4 to 6 [24]. At these times, the excess solution was carefully removed using an automatic pipette, and the tube was weighed again. The swelling index was calculated using Equation (2):(2)$$Swelling\;index\;(\%)=\frac{\left(WS-WD\right)}{WD}\times 100$$
where *WS* is the weight of the film after swelling, and *WD* is the initial weight of the dried film.

For the films’ water resistance, a goniometer (Drop Shape Analysis, DAS 30S model, Kruss; Hamburg, Germany) was applied to measure the contact angle of water droplets on the formulation’s surface. The samples were cut into fragments (2 cm × 1 cm), and 11 μL of distilled water was deposited on the surface of each sample using a micro-syringe. Immediately after applying the water, digital images were captured by a camera, and the contact angle was determined as the angle formed between the line tangent to the droplet at the point of contact and the line drawn along the film’s surface. This measurement was performed using specialized software (model DSA4) within 5 s after the droplet deposition. Both sides of the formulations were evaluated.

### 2.5. Fourier Transformed Infrared (FTIR) Spectroscopy

The samples underwent analysis using FTIR spectroscopy to explore the chemical composition and interactions of the constituents. A Bruker FTIR spectrometer (Alpha-T FTIR model, Billerica, MA, USA) equipped with an Attenuated Total Reflectance (ATR) module was employed to examine the formulation and raw materials of samples. The spectral range from 4000 to 400 cm^−1^ was utilized with a resolution of 4 cm^−1^ and 24 scans. The ATR-FTIR spectra of the samples were leveraged to construct a chemometric model (multivariate analysis) through principal component analysis (PCA).

### 2.6. Chemometrics Analysis—PCA Model

After obtaining the spectrum, an unsupervised chemometric model was developed to differentiate between sample groups, identify key variables, and detect potential outliers [25,26]. Initially, the raw data were pre-processed, which is a crucial step in chemometrics for multiple reasons. The transformation of raw FTIR spectra into cleaner spectra improves the quality of the data used in chemometric analyses, allowing for the development of more robust chemometric models [25,27]. This study tested the following pre-processing techniques individually and in combination to identify the most suitable method for the data: mean center, autoscale, smoothing (Savitzky–Golay), generalized least squares weighting (GLSW), and derivative (Savitzky–Golay).

The principal components were chosen based on their eigenvalues, and a score plot was created to analyze the samples and gather insights into the chemical characteristics and potential interactions within the formulations [25]. The PCA model was developed using MATLAB 7 and PLS-Toolbox 4 from the Eigen-vector Research Group.

### 2.7. Bioadhesive Strength

Both formulations were assessed for bioadhesive strength using a device composed of two balanced arms [25,28]. Samples of porcine ear skin, intact or injured, were used as membrane models donated by a local meat house (Frigorífico Padilha, Guarapuava, PR, Brazil). All membrane samples were cleaned and frozen (−20 °C) until use. The injury was induced by burning the skin on a heating plate at 85 °C for 7 s [29]. The skin (4 cm in diameter) was fixed on a glass plate.

To evaluate the bioadhesive resistance of the hydrogel, 0.8 g of formulation was added in contact with the skin fragment, and a force of 1 N was applied for 60 s. Then, water was carefully introduced through a plastic tube on the opposite side until separation between the skin and the hydrogel was observed. For films, the skin was hydrated with 150 µL of phosphate buffer pH 5.5, and the films were placed in contact with the skin fragment, applying the same force and contact time used to evaluate the hydrogel. After detachment between skin and formulations, the volume of water used was measured using a graduated cylinder. Bioadhesive strength was determined using Equation (3):(3)$$Bioadhesive\;strength\;(dyne/{\mathrm{cm}}^{2})=\frac{\left(V\times G\right)}{A}$$
where *V* is the amount of water (g) required to detach the sample from tissue, *G* is the acceleration of gravity (980 cm/s^2^), and *A* is the area of exposed tissue (cm^2^).

### 2.8. Antioxidant Activity

The formulations’ antioxidant potential was assessed by measuring the radical scavenging capacity of ABTS^+^ [23]. To prepare the ABTS^+^ work solution, a stock solution (7 mM) was mixed with sodium persulfate (140 mM) 12 h before the assay and further diluted with phosphate buffer at pH 7.0, resulting in a final concentration of 42.7 µM ABTS^+^. To test the samples, aliquots of HG-CGG or F-CGG weighing 20.5 mg were exposed to varying volumes of ABTS^+^ radical (1, 2, 4, 6, and 8 mL), obtaining concentrations of 2.56 to 20.5 mg/mL (corresponding to 0.055 to 0.44 mg/mL of CGG for HG and 1.40 to 11.22 mg/mL of CGG for film). Then, ABTS^+^ solution was added to the tubes, mixed on an orbital shaker for 10 min, and incubated at room temperature for 30 min. An ABTS^+^ solution was left under the same reaction conditions and used as a negative control. After incubation, the absorbance of the solution was measured at 734 nm using a spectrophotometer (UV-1800 model; Shimadzu, Kyoto, Japan). Samples of both formulations were dispersed in phosphate buffer pH 7.0 and used as blanks. The percentage of radical elimination was calculated according to Equation (4):(4)$$SC\%=100-\frac{\left((AbsA-AbsB)\times 100\right)}{AbsC}$$

*SC%* is the scavenging capacity measured as a percentage, *AbsA* represents sample absorbance, *AbsB* is the blank absorbance, and *AbsC* represents the negative control absorbance.

### 2.9. Biocompatibility Evaluation

Considering the data derived from the antioxidant assay and bioadhesive potential, the film was selected for further investigation regarding its safety profile. This involved examining its biocompatibility through hemolysis testing and cell viability assessment using the MTT assay. A concentration curve was prepared based on the gum concentration and tailored to meet the specific requirements of each assay.

#### 2.9.1. Hemolysis Assay

The hemolysis assay was conducted using the blood of healthy human volunteers (Research Ethics Committee of the Federal University of Paraná–Brazil, #43948621.7.0000.0102) [30]. Blood was collected by venipuncture in heparinized tubes to induce anticoagulation. It was then centrifuged (1200× *g*/5 min) to yield the red cell fraction and washed with a 0.9% NaCl solution three times to obtain only erythrocytes and remove blood plasma. The erythrocytes were resuspended in the 0.9% NaCl solution at 10% hematocrit (*v*/*v*). Fragments of F-CGG weighing 20.5, 41, and 82 mg (corresponding to 11.22, 22.44, and 44.88 mg of gum, respectively) were placed into microtubes containing 700 µL of 0.9% NaCl solution and equilibrated for approximately one hour. Following this, 100 μL of resuspended erythrocytes was added to the tubes. The positive and negative controls were prepared using distilled water or 0.9% NaCl solution. Additionally, blank samples were prepared, containing the HG-CGG and F-CGG immersed only in the 0.9% NaCl solution instead of the erythrocyte suspension. Finally, the samples were incubated for 1 h at 37 °C, centrifuged (1200× *g*/5 min), and the supernatant’s absorbance was measured in a spectrophotometer (540 nm; UV-1800 model, Shimadzu, Japan). The percentage of hemolysis was calculated according to Equation (5):(5)$$\%\;of\;hemolysis=\frac{\left(AbsA-AbsB\right)}{AbsC}\times 100$$

*AbsA* is the sample absorbance, *AbsB* is the blank absorbance, and *AbC* is the positive control absorbance.

#### 2.9.2. Cell Viability Assay

L-929 cells were kept in culture conditions, 37 °C and 5% CO_2_ (Cell incubator MCO-17 0ACL-PA model; PHC Corporation, Moriguchi, Japan), in Dulbecco’s modified Eagle’s medium (DMEM) supplemented with 10% fetal bovine serum (FBS) and 1% penicillin and streptomycin (10,000 U and 10 mg/mL, respectively). Following this, the cells (1 × 10^5^ cells/mL, equivalent to 1 × 10^4^ cells/well) were seeded in 96-well plates and treated with the F-CGG dispersed in the medium (50–1000 µg/mL of the formulation, corresponding to 27–547 µg/mL of CGG). The medium was used as a negative control. After 24 h under culture conditions, the medium was replaced with 100 µL of MTT solution (0.5 mg/mL in medium) and returned to culture conditions for 3 h. The medium was removed, and the formazan crystals were solubilized in 100 µL of DMSO (dimethyl sulfoxide). The absorbance was read at 570 nm (Cytation 5; Biotek, Winooski, VT, USA). Cell viability was calculated concerning the negative control and expressed as a viability percentage. The assay was performed in triplicate both intra- and inter-experimentally.

### 2.10. Wound-Healing Assay

The L-929 cells (5 × 10^5^ cells/mL) were seeded in a 96-well plate and kept in culture conditions for 24 h for confluency. Afterward, the monolayer was disrupted using a 200 µL micropipette tip. The medium was removed, the wells were washed with PBS, and then they were treated with the F-CGG dispersed in DMEM (50 and 1000 µg of the formulation). A negative control was used for DMEM. Images were acquired immediately after treatment (T0) and every 4 h for 24 h (T4, T8, T12, and T24) in the phase contrast mode at ×10 (Cytation 5; Biotek, Winooski, VT, USA). The assay was performed in triplicate.

### 2.11. Statistical Analyses

The results were obtained in triplicate and are expressed as the mean ± SD. The normality of the data distribution was checked using the Shapiro–Wilk test. Subsequently, a *t*-test or one-way analysis of variance (ANOVA) was applied, followed by Tukey’s post-test, which was based on the experimental design. Significance levels were defined as *p* < 0.05. The statistical software GraphPad Prism^®^ version 8 was used for all statistical analyses and figure creation.

## 3. Results and Discussion

### 3.1. HG Preparation and Characterization

Regarding skin application, especially wound healing, HGs are often preferred over semisolid pharmaceutical forms due to their ease of spreading and removal and pleasant sensorial and organoleptic aspects [4,31,32]. Their high water content also helps reduce pain during application, especially on mucous membranes and injured or burned skin [4]. Additionally, their hydrophilic nature allows them to absorb wound exudate and maintain a moist environment and gas exchange [33,34]. However, the semisolid form must exhibit good gelation properties, resulting in an appropriate apparent viscosity to remain at the application site and effectively reflect their practical application [34,35]. These characteristics depend on various factors, including the polymer profile and concentration [35]. Considering this, we evaluated four gum concentrations during the preformulating step of development to select the optimal one. We visually assessed the formulations to ensure they did not exhibit excessive fluidity or overly high viscosity, which could hinder their applicability in wound healing (Figure 2). After identifying the concentration that provided the best balance of properties, we proceeded with the formal testing of the selected formulation.

The apparent viscosity of the HG increased with the concentration of gum. The semisolid containing 1% (*w*/*w*) CGG exhibited a near-liquid consistency and flowed over the support, suggesting challenges during the application, ease of removal, and low tissue permanence, which may be undesirable (Figure 2A). Conversely, formulations containing 3% (*w*/*w*) (Figure 2C) and 6% (*w*/*w*) (Figure 2D) CGG demonstrated difficulty in being removed from the support, signifying the requirement for higher strength during application and removal, which could pose potential harm and indicate a longer tissue permanence. In contrast, HG containing 2% (*w*/*w*) CGG (HG-CGG) displayed an adequate strength and satisfactory viscosity, making it the most suitable sample and the chosen continuation for our work (Figure 2B).

The 2% *w*/*v* HG-CGG had a pH of 7.57 ± 0.07, consistent with other studies where gum-based semisolid formulations for cutaneous application were developed [36,37,38]. Formulations with an acidic or basic pH can harm the skin, and those with a moderate pH may be more suitable for preventing tissue damage in cutaneous injuries [39]. Therefore, a neutral range of pH may be compatible with chronic skin injuries, such as wounds [40].

The physical stability was determined by submitting the formulations to centrifugal force. Centrifugation stresses the samples by causing an increase in particle mobility, which can potentially lead to instability of the emulsions and semisolid formulations [19,37]. After the test, the HG-CGG did not exhibit any macroscope instability, such as a break of polymeric chains or precipitation, suggesting physical stability even under forced conditions.

In semisolid formulations, it is essential to determine spreadability and viscosity, to ensure proper skin application and storage. Spreadability is the ability of the formulation to spread on a surface. Additionally, this parameter can determine how the formulation can expand when a force is applied, indicating the force required to spread the formulation on the skin, and is related to the proper transfer of the dose to the application area, which is crucial to determining product acceptability [36,38,41]. Based on our findings, HG-CGG exhibited an Sf of 2.39 ± 0.123 mm^2^/g. In comparison, other studies have reported different results for gum-based hydrogels: locust gum hydrogels showed an Sf of around 3.4 mm^2^/g [37], gellan gum hydrogels around 4.5 mm^2^/g [28], and xanthan gum hydrogels about 7.6 mm^2^/g [38]. These variations indicate that CGG hydrogels demonstrate a high level of consistency and may be well-suited for specific applications, such as wound coverage. Despite this, the area of spreadability increased as the weight accumulation increased, as shown in Figure 3A. This indicates that the product is easily spreadable and may provide adequate skin coverage without causing pain during application at the wound site.

Dynamic viscosity is the resistance that occurs when these layers move against each other and is a critical property of semisolid preparation [42]. The viscosity property is associated with formulation consistency and residence time in the wound site. At the lower shear rate (0.034 s^−1^), the obtained viscosity was 5699 ± 198 mPas, while at the highest shear rate (0.77 s^−1^), the viscosity was 2169 ± 46 mPas. The viscosity decreases when the share rate increases, indicating that the HG has a non-Newtonian flow. Still, this behavior could suggest a pseudoplastic flow (Figure 3B), which is desired in pharmaceutical formulations for topical application because it ensures proper spreadability of the dispersed phase under low shear stress, particularly to treat wounds, and increases the residence time and contact with the injured tissue [31,38,43]. Moreover, this behavior has already been demonstrated in other studies that evaluated hydrogels based on natural gum [38,43].

### 3.2. Film Preparation and Characterization

Developing films involves determining the optimal concentration of film-forming polymers and plasticizers. Figure 4 shows representative images of the films in various stages of development. Some studies have shown the use of CGG as a polymeric material for films at concentrations between 0.5% and 2% (*w*/*v*) [17,44,45]. Thus, initial concentrations of 1% and 2% (*w*/*v*) CGG were tested, showing the formation of very thin and fragile films (Figure 4A,B). Based on this, higher gum concentrations (4% and 6%, *w*/*v*) were tested, revealing that the 6% (*w*/*v*) concentration was sufficient to allow the film to detach from the drying surface without breaking (Figure 4D). The amount of glycerol was also evaluated (2%, 4%, and 5%, *w*/*v*) revealing that the concentration of 5% (*w*/*v*) resulted in visually flexible films, which are desirable for cutaneous application. Optimized F-CGG presented a homogeneous, transparent appearance with a yellow color (Figure 4D), consistent with other studies that used CGG to obtain films [44,45].

Cutaneous-friendly films are desired to be as thin as possible to ensure comfortable application [46]. The F-CGG had approximately 0.6 mm of thickness (Table 1), considered thin according to the literature [11]. Other films intended for cutaneous use have also demonstrated similar thickness values [47,48]. In the transparency test, the F-CGG presented transmittance values of around 52% (Table 1), which indicates that it can be considered transparent [21]. Furthermore, the scan spectrum is shown in Figure 4E. This feature is particularly valuable in the clinical context, especially for treating critical skin injuries such as severe burns and painful wounds, and post-operative care, as it allows for the visualization of the healing process without physically removing the film. Thus, this transparency facilitates continuous monitoring of the evolution of the injury, reducing the risk of infection or rupture of the recovering tissue [23,49].

The average weight of F-CGG was 82.03 ± 1.91 mg/cm^2^ (Table 1), indicating reduced mass variation based on SD. The dose uniformity of polymeric films is directly related to weight uniformity. Films with reduced weight variation typically show consistent contents [23,30]. Therefore, the uniform weight of F-CGG suggests that consistent therapeutic doses may be achieved when an active substance is added.

Figure 5 displays the images obtained by SEM microscopy from F-CGG. As can be observed, the surface of F-CGG was uniform with minimal irregularities and no porosity (Figure 5A–C). The cross-section shows a dense and compact morphology, similar to those already reported for other films made of CGG (Figure 5D) [44,50]. The uniformity and dense appearance may be due to hydrogen bonds between the polymer chains, bringing them together in a cohesive and stable fashion [50].

Table 1 presents the mechanical properties of films. Films should preferably be flexible for skin application and adapt comfortably to the application surface, especially at injured sites [51]. The F-CGG film demonstrated high flexibility, remaining intact after 300 folds in the folding endurance test, indicating that it can form properly on the skin without breaking or forming grooves [11,21]. Flexibility is crucial to ensure the film can be applied to flexural areas without losing its structural properties [52].

Other critical mechanical parameters for films include maximum stress, elongation, and Young’s modulus, which showed values of 1.16 ± 0.24 MPa, 40.38 ± 5.00%, and 2.96 ± 0.87 MPa, respectively (Table 1). Maximum tension represents the maximum stress supported by the film before its rupture, reflecting the strength of the material, while elongation indicates the film’s elasticity [53]. F-CGG presented maximum tension and elongation parameters similar to other studies where films for cutaneous use were developed [30,50,54]. Young’s modulus indicates the film’s resistance to deformation, and it is important for it to be compatible with the skin’s resistance to deformation. Studies have shown that Young’s modulus of the skin varies between 0.02 and 57 MPa, depending on factors such as hydration, thickness, and type of force applied. This indicates that F-CGG demonstrated tissue-compatible results cutaneously [55]. Films with lower Young’s moduli, such as F-CGG, indicate greater flexibility, essential for application to injured areas requiring materials that follow skin movements without breaking [56,57]. These results suggest that F-CGG has an adequate combination of flexibility and strength, meaning it is potentially effective for use in skin lesion treatments.

The evaluation of the contact angle gives information about the hydrophilicity degree of the film surface, while the swelling test identifies its ability to absorb water from the environment. Both parameters are essential to investigate the interaction of the formulation with biological fluids [52]. Of particular importance, the quaternary ammonium group insertion into the CGG structure increases the hydroxyl group density, leading to increased gum solubility given the enhanced supramolecular interactions with polar groups [58]. The findings demonstrated high swelling potential for F-CGG, presenting an index of 2032 ± 211% after 6 h of experimentation, and maintaining similar values up to 24 h of testing (*p* > 0.05; Figure 6A). The high swelling index suggests that the films can effectively absorb aqueous biological fluids on the skin surface, such as wound exudate, and minor bleeding.

The contact angle can indicate whether a surface is hydrophilic (less than 90°) or hydrophobic (greater than 90°) [11]. It was observed that F-CGG presents different characteristics depending on the face evaluated, being hydrophobic on the upper side, which is exposed to air during drying, and hydrophilic on the lower side, which encounters the Petri dish’s surface during drying (Figure 6B, *** *p* < 0.001). Both behaviors can be found in the literature for cutaneous films and can provide distinct advantages. In this study, we evaluated the swelling behavior of the CGG-based films, which measures the amount of water absorbed and retained by the material over specific time intervals. The result underscores the hydrophilic nature of the CGG-based films. While the contact angle measurements indicated different hydrophilic properties between the upper and lower surfaces of the film, the swelling behavior offered more robust indication on the material’s interaction with water, particularly relevant for wound-healing applications where moisture retention is critical.

Hydrophilic films are designed to absorb moisture from the wound bed and create an environment less conducive to microbial contamination [30,59,60]. On the other hand, hydrophobic films can promote an increase in the adhesion of fibroblasts, benefiting the wound-healing process [60]. Therefore, depending on the formulation’s desired therapeutic objectives, both sides of the F-CGG can be selected for application with direct skin contact.

### 3.3. FTIR Spectroscopy

Figure 6 depicts the spectra recorded for the formulations (hydrogel and film) and guar gum. For bulk CGG (Figure 7A), stretching bands of O–H bonds of hydroxyl groups were observed around 3305 cm^−1^. These bands are characteristic of phenols and carbohydrate monomers, including mannose and galactose [61]. The bands around 2900–2800 cm^−1^ are due to the asymmetric stretching of the C–H bond of –CH_2_ groups and the symmetric stretching of C–H of –CH_3_ groups of the sugar structure, respectively. The band at 1014 cm^−1^ is ascribed to stretching the C–O bond from –COCH_3_ groups. The sharp bands at 1654 cm^−1^ and 1459 cm^−1^ depict the presence of quaternary ammonium cations. Finally, the band at 1371 cm^−1^ refers to C–N bonds [44].

In the ATR-FTIR spectrum of the film, the characteristic bands for CGG were identified. Bands attributable to O–H and C–O stretching were observed at 3292 and 1027 cm^−1^, respectively. Some bands observed in this spectrum allowed the identification of glycerin groups, as shown in Appendix A. Carbonyl groups belonging to aldehydes, primarily acrolein, can be identified by the characteristic bands around 1650–1800 cm^−1^ resulting from the dehydration of glycerin [25]. The bands referring to the quaternary ammonium cations were still present, as can be seen in Figure 7B. This may suggest that the film preparation processes did not modify the chemical structure of CGG, potentially influencing the bioadhesion results by enhancing this feature given the positive charge of the biopolymer. Regarding the HG, a distinct spectrum profile was observed. The broad band observed around 3278 cm^−1^ is for O–H stretching of the CGG (hydroxyl groups) and probably the water in the formulation. The band at 1636 cm^−1^ can be ascribed to the quaternary ammonium cations. The difference in gum concentration could explain the main similarity between the film and CGG spectra and the few peaks observed in the HG profile. In the film formulation, the CGG is three times more concentrated than in the hydrogel, which could result in less pronounced bands.

Analyzing chemical changes using ATR-FTIR spectra is complex. With its wide range of wavenumbers (4000–400 cm^−1^), identifying subtle alterations in the composition can be challenging [25]. In this context, a chemometric model plays a significant role in interpreting spectra and determining the underlying variables. The PCA model serves as an exploratory approach to reducing data dimensionality, pinpointing principal variables and outliers, and facilitating their classification [25,26]. It has been applied in formulation development to understand the interactions between excipients [25,30]. Figure 8A demonstrates that two principal components were found in the ATR-FTIR spectra of the formulations, which were selected using the eigenvalue criterion. After applying the pre-processing smoothing and GLSW, it was possible to observe that these two components are responsible for 100% of the cumulative variance in the data. As observed in the ATR-FTIR spectra, the score graph shows that the CGG and film formulation have chemical similarities once they are in the same plan of PC1, which is responsible for 90.03% of the cumulative variance (Figure 8B). On the other hand, the HG is chemically dissimilar to the F-CGG, which could be due to the preservative and higher water content.

### 3.4. Bioadhesive Strength

The bioadhesive behavior of formulations is shown in Figure 9. For the semisolid form, the bioadhesive force was 7282.378 ± 218.337 dyne/cm^2^ for intact skin and 4612.17 ± 261.81 dyne/cm^2^ for burned skin (*p* < 0.001). This finding suggests that HG-CGG has a reduced adhesion potential when the skin loses its integrity. Barbalho and co-workers reported that contact with a hot plate for 7 s resulted in a noticeable reduction in the thickness of the epidermal layer, increased transepidermal water loss, and deformation of epithelial cells, similar to a second-degree burn injury [29]. Although other changes can also impact the bioadhesive potential, the increase in transepidermal water loss represents a significant challenge. CGG is a highly hydrophilic gum that depends on water molecules to relax its polymer chains. As a result of the water loss, CGG may not interact efficiently with the tissue, reducing the bioadhesive strength of the formulation [21].

In contrast, F-CGG bioadhesive strength in uninjured skin was 5662.85 ± 501.63 dyne/cm^2^, and it was 6554.14 ± 540.57 dyne/cm^2^ for injured skin (*p* > 0.05). This means that the adhesiveness of the developed films remains consistent regardless of the integrity of the skin, which is beneficial as it ensures that the film stays in place even when applied to wounds. Additionally, the transparency of F-CGG allows for continuous monitoring of the wound without needing to remove the film, potentially reducing the need for frequent dressing changes [49,59]. Bioadhesion can occur through various mechanisms, including hydrogen bonds and van der Waals forces. Gums contain several hydrophilic functional groups, such as hydroxyl and carboxyl, and in the case of CGG, quaternary ammonium groups in their chemical structure. These groups allow gums to form interactions with biological membranes [25,62].

Therefore, a significant statistical difference was observed between HG-CGG and F-CGG in both injured and non-injured groups. The HG-CGG formulations demonstrated the highest bioadhesive strength on intact skin, preventing water loss and allowing for the interaction of polymer chains with the tissue. Conversely, F-CGG showed a higher bioadhesive potential on injured skin than HG-CGG, indicating its potential as a cutaneous wound-healing device.

### 3.5. Antioxidant Activity

During the initial healing phase, defense cells release reactive substances to protect the tissue from hazardous agents and stimulate angiogenesis. While these substances have important roles, too many can upset the body’s natural balance. This imbalance can lead to the breakdown of large molecules and harm cell structures, slowing the healing process [4,63]. Therefore, substances with antioxidant properties can play a crucial role in modulating effective wound healing.

The antioxidant activity of the formulations was tested using ABTS^+^ (Figure 10). Overall, F-CGG presented superior antioxidant action to HG-CGG, mainly when higher quantities of the forms were tested (5.13, 10.25, and 20.50 mg/mL; *p* < 0.05). This finding may be related to the superior CGG concentration in the solid formulation. The higher CGG concentration in the F-CGG formulation likely contributes to its increased efficacy in neutralizing free radicals, a critical factor in wound healing and tissue regeneration. The antioxidant properties can be crucial in reducing oxidative stress at the wound site, promoting faster and more efficient healing. Natural gums have desirable properties, including antioxidant activity [64], which was previously assigned to the presence of oligomers in their chemical structure [63,65]. This superiority is likely due to the inherent antioxidant properties of natural gums, which are beneficial for wound-healing applications. The hydroxyl and quaternary ammonium groups on the CGG chains exhibit inherent antioxidant properties, enabling the regulation of the inflammatory response [66]. This characteristic, coupled with the biocompatibility and biodegradability of CGG, makes F-CGG a promising candidate for advanced wound care solutions.

The differences in performance between HG-CGG and F-CGG formulations are primarily due to their compositional variations and preparation steps, which were characterized through several analytical techniques. This allowed us to better understand their behavior in wound-healing applications. HG-CGG is a semisolid formulation with a high water content, characterized by a porous three-dimensional network. This was assessed using rheological measurements and spreadability tests. These analyses demonstrated the hydrogel’s greater flexibility and ease of application across larger surface areas. However, the high water content contributes to reduced bioadhesive strength, especially on injured skin, where the increased transepidermal water loss limits the hydrogel’s ability to interact effectively with the tissue. This was confirmed by our bioadhesion tests using porcine skin models.

In contrast, F-CGG, a solid form produced through solvent casting, was characterized using mechanical property tests, swelling capacity measurements, and contact angle analysis. These methods highlighted its denser and more compact structure, which directly contributes to superior bioadhesive strength, particularly on injured skin. Unlike HG-CGG, the film’s structure allows it to maintain contact under conditions of significant water loss. Additionally, F-CGG exhibited higher mechanical strength and a greater swelling index, making it more effective at absorbing wound exudate while maintaining its integrity. The increased concentration of CGG in the film further enhances its antioxidant activity, as more quaternary ammonium groups are available to neutralize free radicals, as demonstrated through the ABTS radical scavenging assay.

While these findings provide valuable insights into the mechanisms behind the differences in performance between HG-CGG and F-CGG, we acknowledge that there are limitations to the possible extrapolations and hypotheses that can be made based on our results. The in vitro nature of our experiments, including bioadhesion, mechanical strength, and antioxidant activity assays, may not fully capture the complexities of in vivo wound-healing environments. However, the evidence gathered from our experiments suggests that the compositional differences between HG-CGG and F-CGG play a key role in their distinct performances.

### 3.6. Safety Assays

The hemolysis test revealed that exposure of erythrocytes to the formulation resulted in a measurable level of hemolysis, particularly at higher film concentrations (Figure 11A). Although the observed hemolysis increased with larger quantities of the film, it remained below the critical 10% threshold, indicating a low hemolytic potential [67]. However, the presence of hemolysis, even at these low levels, warrants further consideration of the formulation’s components, particularly the cationic charge of the CGG. The cationic charge of CGG could play a role in the interaction with cell membranes, particularly erythrocytes. Cationic polymers interact with negatively charged components of cell membranes, which can lead to membrane disruption and subsequent hemolysis [68,69]. While the hemolysis observed in this study was minimal, the cationic nature of CGG might contribute to this effect by causing slight perturbations in the erythrocyte membrane integrity. Although not severe, this interaction may partly explain the hemolysis detected at higher film concentrations.

Despite this, the overall hemolytic activity of the formulation was low, suggesting that the cationic charge does not significantly compromise erythrocyte integrity under the tested conditions. The cytotoxicity test on L-929 cells (fibroblasts) further supported this, as the formulation did not reduce cell viability, indicating that its cationic charge does not pose a substantial risk to cell health [70] (Figure 11B). This is consistent with the known biocompatibility of natural gums like CGG, which are widely used in various industries due to their safety and low toxicity. In summary, while the cationic charge of CGG might contribute to the minor hemolysis observed, the formulation still demonstrates a favorable safety profile with low hemolytic potential and no cytotoxic effects [6,8]. These findings, combined with the biocompatibility of CGG, suggest that the film formulation is a promising candidate for wound dressing applications, where it can safely support the healing process without causing significant harm to surrounding tissues.

### 3.7. Wound-Healing Assay

The wound-healing process includes cell proliferation to reduce the lesioned area and promote reepithelization [3]. Therefore, considering the lack of cytotoxicity towards L-929, the lower and higher concentrations evaluated in the cell viability assay were analyzed regarding their effect on cell proliferation (Figure 12).

Noticeably, both concentrations promoted cell proliferation compared to the control (DMEM) within 24 h of treatment, almost closing the cell wound. Other matrices with guar gum were able to increase the proliferation of fibroblasts [71,72], possibly because of the degradation products of the biopolymer in the medium. Recent studies also demonstrated the potential healing properties of CGG using other models, such as the in vivo *Staphylococcus aureus*-infected wound-healing assay [73]. To support the healing process, it is important to establish a moist environment that facilitates wound healing. This is achieved by eliminating harmful bacteria, regulating the local inflammatory response, and expediting collagen deposition while promoting blood vessel regeneration [73].

The results highlight the potential of CGG as an effective agent in enhancing wound healing, mainly through its ability to promote cell proliferation. The significant increase in fibroblast cell proliferation observed within 24 h of treatment, especially compared to the control, underscores its potential to accelerate wound closure. This rapid cellular response suggests that CGG could play a crucial role in the initial stages of wound healing, where restoring the damaged tissue is critical. The biopolymer’s non-cytotoxic nature and potential antioxidant action further enhance its appeal, making it a viable option for wound care applications.

When considering the use of CGG for wound healing, it is important to highlight its advantages and limitations compared to traditional wound-healing devices and synthetic hydrogels. CGG is notable for its strong bioadhesive properties due to the presence of quaternary ammonium groups. These properties ensure prolonged contact with the wound site, creating a moist environment that is critical for tissue regeneration. Additionally, CGG-based hydrogels and films demonstrate exceptional hydrophilicity, with a high swelling capacity that allows them to absorb large volumes of wound exudate while maintaining moisture. This sets CGG apart from synthetic hydrogels, which may not achieve the same level of fluid retention or bioactivity. Furthermore, CGG is a naturally biocompatible and biodegradable material, ensuring a safer interaction with biological tissues without the cytotoxicity or removal difficulties associated with synthetic alternatives. However, CGG has a lower mechanical strength, making it less suitable for wounds in areas under frequent movement or stress, where synthetic hydrogels or silicone-based dressings might be more durable. Additionally, CGG hydrogels have shown reduced bioadhesion on injured skin, indicating the need for further formulation optimization. Nonetheless, CGG’s multifunctionality, including its antioxidant activity, positions it as a promising material for wound-healing applications, offering unique benefits compared to conventional and synthetic alternatives.

Another relevant aspect to discuss is that this study presents an initial exploration of the potential of CGG in hydrogel and film formulations for wound-healing applications. Our results indicate promising short-term stability, particularly in terms of swelling behavior over a 24 h period (F-CGG) and the centrifugation test (HG-CGG). However, further research is required to evaluate the long-term stability, especially with extended exposure to water. The absence of cross-linking agents in our formulations may result in material degradation after prolonged contact with water, leading to a reduction in mechanical integrity over time. Moreover, microbiological stability is crucial for wound-healing applications, and future studies should concentrate on improving the formulation by integrating preservatives to prevent microbial contamination. These observations suggest that incorporating cross-linking agents and preservatives could greatly enhance the durability and clinical suitability of CGG-based materials. In future investigations, our goal is to integrate active substances to potentiate the per se wound-healing potential identified to CGG and substances that can serve dual functions, such as acting as plasticizers to films while also offering pharmacological effects. For example, incorporating aloe vera, as demonstrated in the study by Alves and collaborators (2024) [25], could potentially enhance the wound-healing properties of the formulation. This strategic approach will enable us to maximize the therapeutic potential of the hydrogel while upholding the desired mechanical and bioadhesive properties for effective wound care. By addressing these aspects, future research will help unlock the full potential of CGG in therapeutic wound care.

Lastly, the decision to investigate CGG in both hydrogel and film forms was driven by the specific benefits each formulation offers for wound-healing applications. Hydrogels, with their high water content and flexibility, are well-suited for maintaining a moist environment and covering irregular wound surfaces, while films provide a superior mechanical strength, longer adhesion, and better control over the release of therapeutic agents. Our findings indicate that the film form of CGG shows enhanced bioadhesion, mechanical properties, and antioxidant activity, suggesting that CGG may be more beneficial for the preparation of solid, long-lasting wound care materials. This comprehensive approach not only underscores the versatility of CGG but also highlights its potential for various therapeutic applications. As the first study to explore CGG in both hydrogel and film forms, our work provides valuable insights that can lay the groundwork for future research, showcasing the wide-ranging applicability of natural gums in the development of innovative formulations for wound healing.

## 4. Conclusions

This study successfully developed and characterized CGG-based HGs and films, demonstrating their potential as innovative wound-healing devices. Both formulations demonstrated suitable general physicochemical properties for skin application (viscosity, spreadability, pH, mechanical features, and physical stability). Remarkably, the HG and F had distinct bioadhesive profiles for normal (HG > F) and lesioned skin (F > HG) and antioxidant potential (F > HG).

Importantly, F-CGG maintained its bioadhesive properties regardless of the integrity of the skin, which is advantageous for ensuring that the dressing remains in place on wounds. This feature, combined with F-CGG’s transparency, can allow continuous wound monitoring, potentially reducing the need for frequent dressing changes and thus minimizing patient discomfort. Safety assessment suggested F-CGG biocompatibility. Finally, the wound-healing assay showed that F-CGG promoted cell proliferation, which may support re-epithelialization, wound closure, and its applications as a wound-healing device. However, more research is needed to better comprehend its full potential, including stability assessments, tests of the antimicrobial action, and in vivo studies, which are required to evaluate the efficacy, safety profiles, and potential clinical applications.

## Figures and Tables

**Figure 1 pharmaceutics-16-01233-f001:**
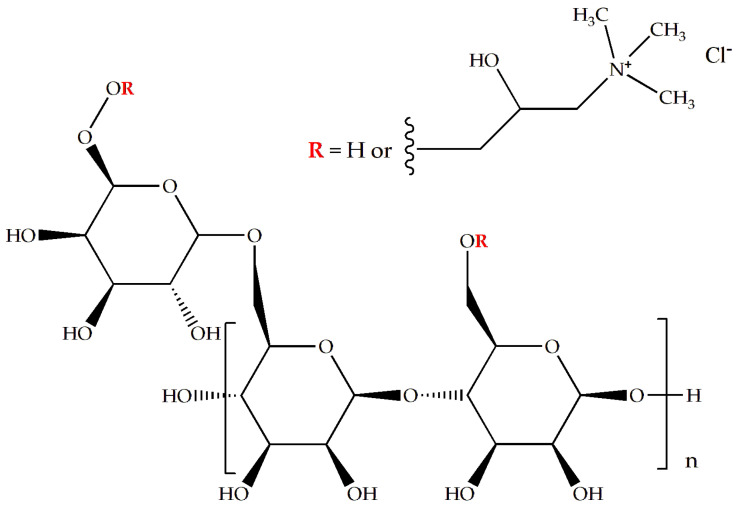
Chemical structure of cationic guar gum (CGG).

**Figure 2 pharmaceutics-16-01233-f002:**
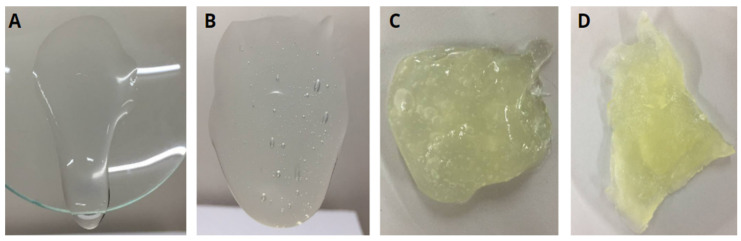
Macroscopic appearance of hydrogels during development formulation tests at different concentrations of CGG. Hydrogels at 1% (**A**); 2% (**B**); 3% (**C**); and 6% (**D**) *w*/*w* of CGG.

**Figure 3 pharmaceutics-16-01233-f003:**
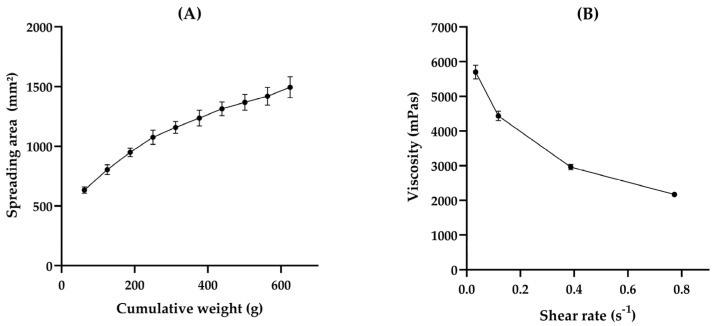
HG-CGG spreadability profile (**A**) and viscogram (**B**).

**Figure 4 pharmaceutics-16-01233-f004:**
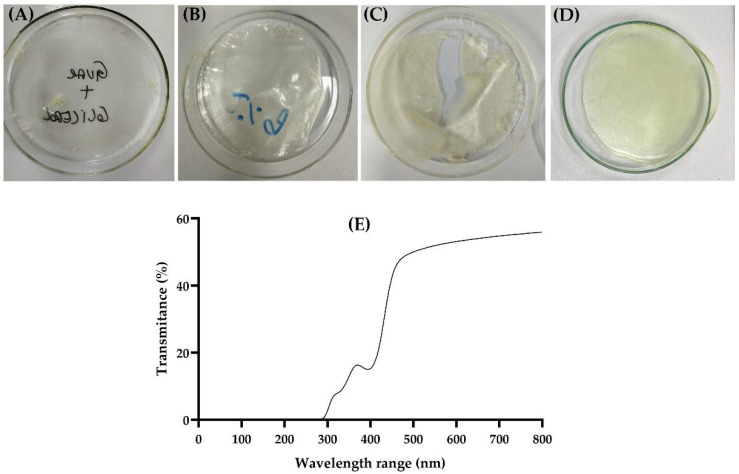
Images of films prepared with 1% (**A**), 2% (**B**), 4% (**C**), and 6% (**D**) (*w*/*v*) CGG during development tests and scans obtained in UV-Vis spectrum to F-CGG (6% *w*/*v*) (**E**).

**Figure 5 pharmaceutics-16-01233-f005:**
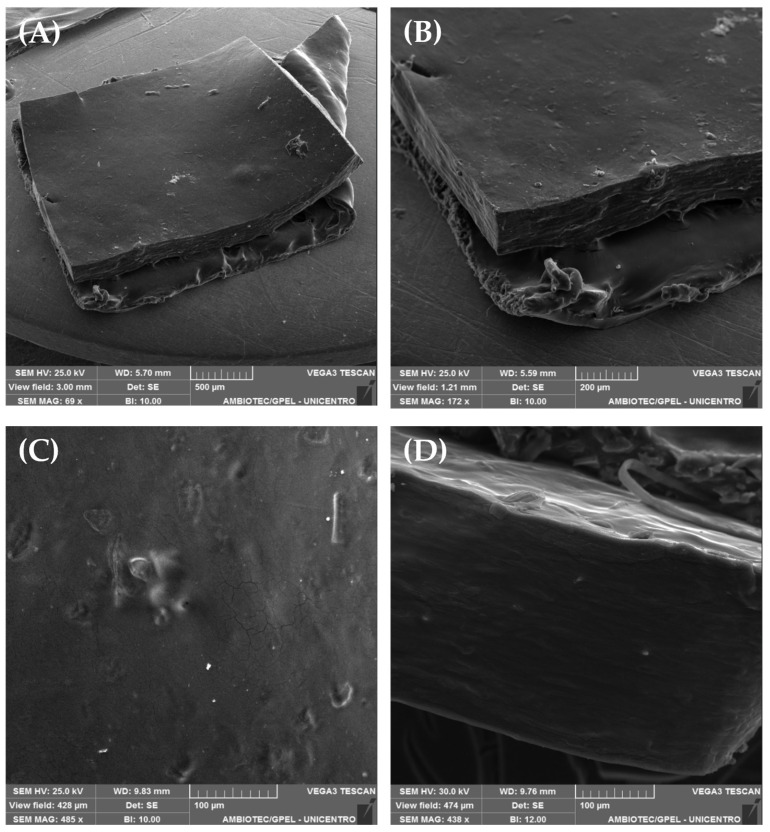
Scanning electron microscopy (SEM) images obtained from the surface portion of the F-CGG film using different magnifications (**A**–**C**) and SEM images obtained from the transversal section of F-CGG after cryofracture (**D**) (scale bars: (**A**) 500 µm, (**B**) 200 µm, (**C**) 100 µm, and (**D**) 100 µm).

**Figure 6 pharmaceutics-16-01233-f006:**
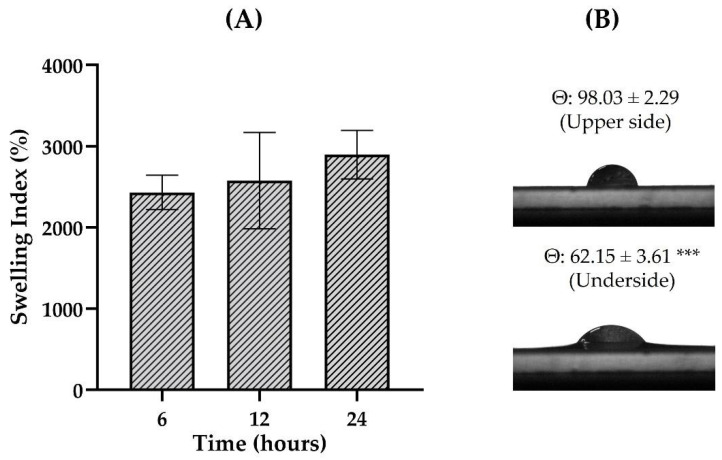
Swelling index (**A**) and representative images of contact angle (**B**) determination evaluated for F-CGG (*n* = 3). The results are expressed as mean ± SD (*n* = 3). Unpaired *t*-test. *p* < 0.001 (***): significant difference between the contact angle obtained on the upper and lower faces of the F-CGG.

**Figure 7 pharmaceutics-16-01233-f007:**
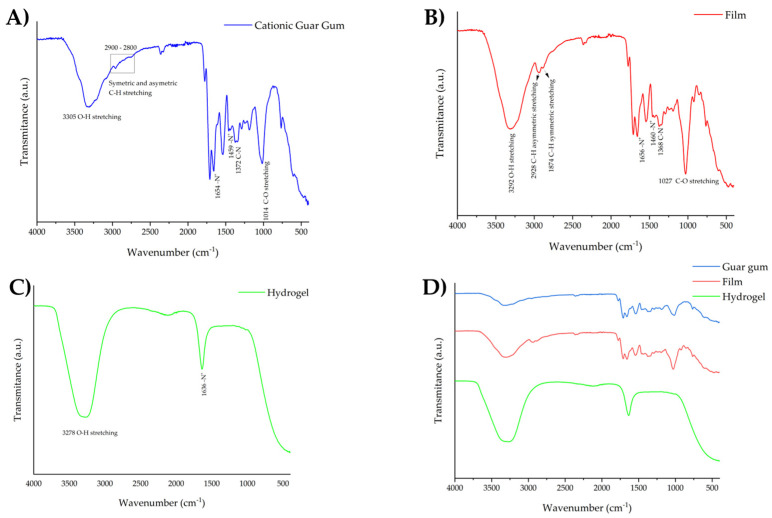
ATR-FTIR of the pure CGG (**A**), F-CGG (**B**), and HG (**C**) and a comparison of them (**D**).

**Figure 8 pharmaceutics-16-01233-f008:**
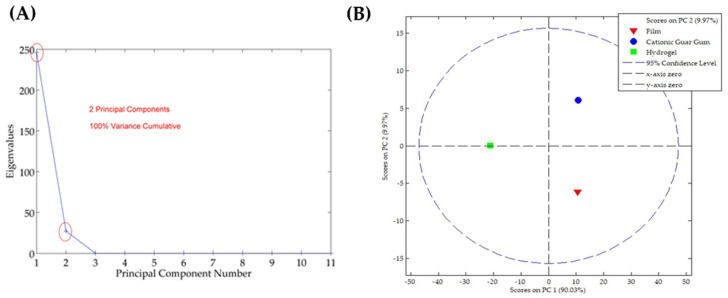
PCA model applied to the ATR-FTIR spectra. (**A**) Eigenvalues against the number of principal components, and (**B**) score plot.

**Figure 9 pharmaceutics-16-01233-f009:**
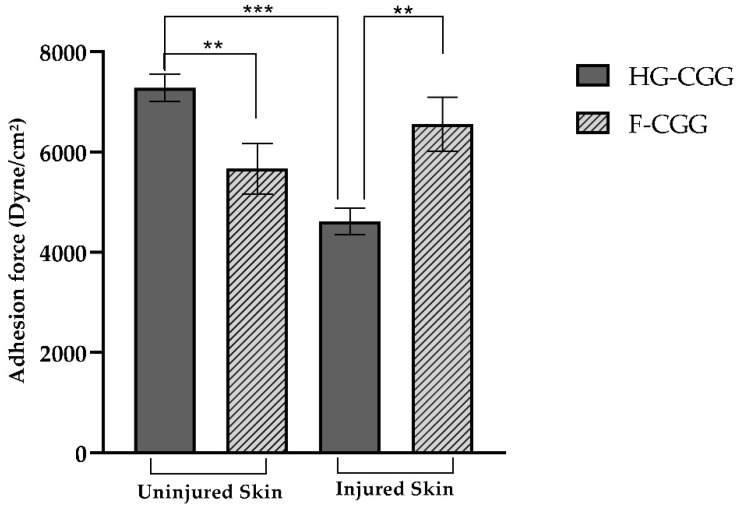
Bioadhesive strength of HG-CGG and F-CGG in uninjured and injured skin. Mean ± SD (*n* = 3). One-way ANOVA followed by Tukey’s test: (***) difference between HG-CGG in uninjured and injured skin (*p* < 0.001); (**) difference between HG-CGG and F-CGG in both uninjured and injured skin (*p* < 0.05).

**Figure 10 pharmaceutics-16-01233-f010:**
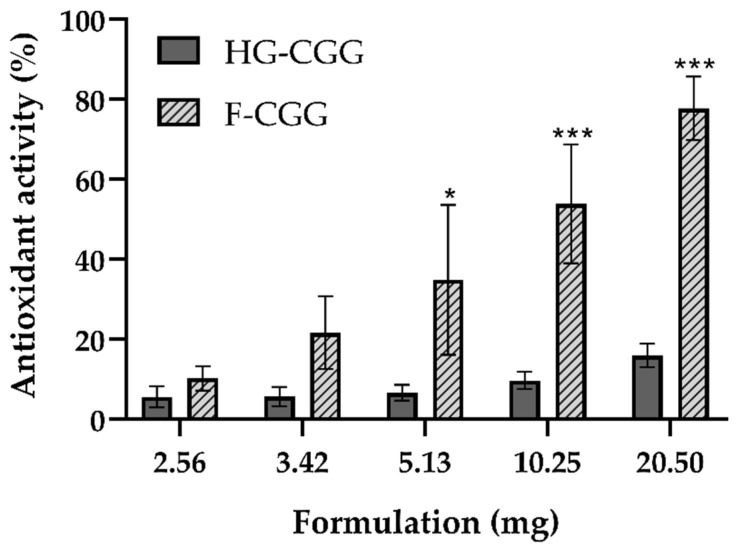
Antioxidant activity of HG-CGG and F-CGG. Mean ± SD (*n* = 3). One-way ANOVA followed by Tukey’s test: (*) difference between HG-CGG and F-CGG in 5.13 mg/mL (*p* < 0.05); (***) difference between HG-CGG and F-CGG in 10.25 and 20.50 mg/mL (*p* < 0.001). For this assay, formulations were weighed (2.56 to 20.5 mg/mL of the formulations), corresponding to a CGG concentration range of 0.055 to 0.44 mg/mL to HG and 1.40 to 11.22 mg/mL to film.

**Figure 11 pharmaceutics-16-01233-f011:**
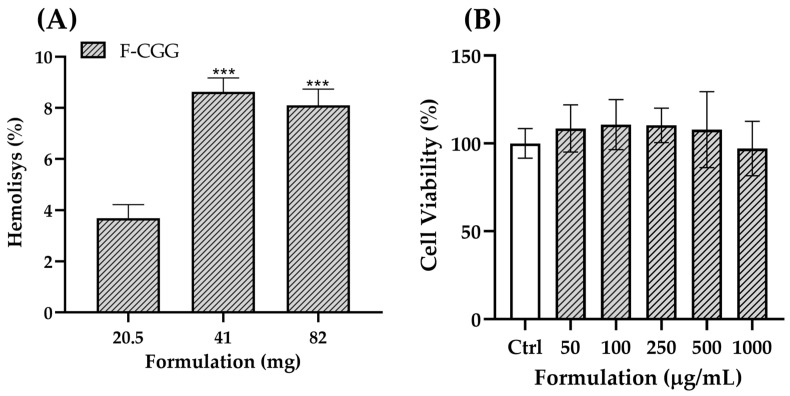
Hemolytic effect (**A**) and cell viability of L-929 cells after 24 h of treatment with F-CGG (**B**). Mean ± SD (*n* = 3). One-way ANOVA followed by Tukey’s test: (***) difference compared to the 20.5 mg group. For the assay in (**A**), film fragments were weighed (20.5, 41, and 82 mg of formulation/tube), corresponding to the CGG concentration ranges of 11.22, 22.44, and 44.88 mg/tube. In (**B**), the range of 50–1000 µg/mL of the formulation corresponded to 27–547 µg/mL of gum.

**Figure 12 pharmaceutics-16-01233-f012:**
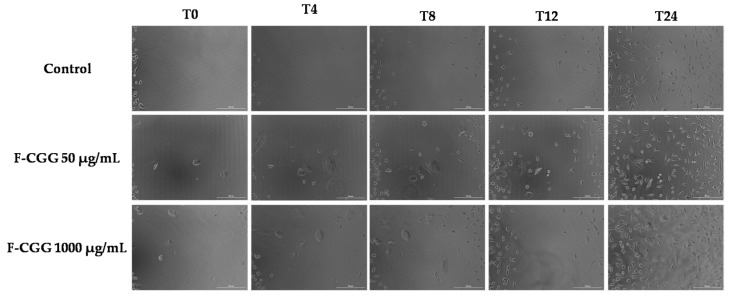
Wound-healing properties of 50 and 1000 μg/mL on L-929 cells (fibroblasts) at 0, 4, 8, 12, and 24 h after the treatment (T; time). DMEM was used as a control. Pictures were taken in the phase contrast mode, 10×, scale bar = 200 μm.

**Table 1 pharmaceutics-16-01233-t001:** Assessment of the thickness, folding resistance, transparency, weight uniformity, and mechanical properties of the F-CGG.

Parameter	Value
Thickness (mm)	0.63 ± 0.07
Transparency (%)	52.46 ± 3.29
Weight uniformity (mg/cm^2^)	82.03 ± 1.91
Folding endurance	>300
Tensile strength (MPa)	1.16 ± 0.24
Elongation (%)	40.38 ± 5.00
Young’s modulus (MPa)	2.96 ± 0.87

## Data Availability

Data is contained within the article or Appendix A.

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
