# Peer review of "Exploring Cationic Guar Gum: Innovative Hydrogels and Films for Enhanced Wound Healing"

_pharmaceutics, 2024, doi:10.3390/pharmaceutics16091233_

Round 1
Reviewer 1 Report
Comments and Suggestions for Authors
1. It looks more like an experimental report. The abstract format does not conform to the conventional writing style of academic papers. And it cannot be just a general statement, there should be data to support the conclusions.
2. What is the chemical structure of Cationic Guar Gum?
3. Which UV spectrum is shown in Figure 3E
4. Figure 5B should illustrate the time for the dry film to be wetted by water. CGG should be hydrophilic in nature, but the wetting time is not enough?
5. Figure 6C, is it necessary to keep from water during infrared testing?
6. The mechanism behind the differences in performance between HC-CGG and F-CCG needs further clarification?
7. As the application of wound healing, what are the advantages and disadvantages of CCG compared with other types of hydrogels or membranes?
Comments on the Quality of English LanguageGood.
Author Response
Reviewer 1
- It looks more like an experimental report. The abstract format does not conform to the conventional writing style of academic papers. And it cannot be just a general statement, there should be data to support the conclusions.
Answer: Thank you for your valuable feedback. We acknowledge the need to enhance the abstract to align with the conventional academic paper format. As a result, we have revised the abstract to ensure that it provides a general overview and includes specific data to substantiate the conclusions. We have incorporated quantitative information about the main findings, such as the developed materials' bioadhesive properties, antioxidant potential, and biocompatibility.
To specify, we have included the following data in the abstract:
- The bioadhesion of the film was 6554.14 ± 540.57 dyne/cm² on injured skin, demonstrating its effectiveness in maintaining adherence to the wound site.
- The film's swelling index reached 2032 ± 211% after 6 hours, highlighting its ability to absorb biological fluids.
- The film exhibited superior antioxidant capacity compared to the hydrogel, with an ABTS+ radical scavenging activity of 20.50 mg/mL.
We trust that these changes will align the abstract with academic expectations and appreciate constructive suggestions. All the modifications are highlighted in green in the revised version of the manuscript. The revised abstract is available below:
This study developed and characterized hydrogels (HG-CGG) and films (F-CGG) based on cationic guar gum (CGG) for application in wound healing. HG-CGG (2% w/v) was prepared by gum thickening and evaluated for pH, stability, spreadability, and viscosity. F-CGG was obtained using an aqueous dispersion of CGG (6% w/v) using the solvent casting method. F-CGG was characterized for thickness, weight uniformity, morphology, mechanical properties, hydrophilicity, and swelling potential. Both formulations were evaluated for bioadhesive potential on intact and injured porcine skin, as well as antioxidant activity. F-CGG was further studied for biocompatibility using hemolysis and cell viability assays (L929 fibroblasts), and its wound healing potential by the scratch assay. HG-CGG showed adequate viscosity and spreadability profiles for wound coverage, but its bioadhesive strength was reduced on injured skin. In contrast, F-CGG maintained consistent bioadhesive strength regardless of skin condition (6554.14 ± 540.57 dyne/cm² on injured skin), presenting proper mechanical properties for skin application (flexible, transparent, thin, and resistant) and high swelling capacity (2032 ± 211% after 6 hours). F-CGG demonstrated superior antioxidant potential compared to HG-CGG (20.50 mg/mL ABTS+ radical scavenging activity), in addition to being biocompatible, exhibiting low hemolytic potential and no cytotoxicity to fibroblasts. F-CGG promoted a significant increase in fibroblast proliferation, leading to wound area reduction by 50% (12 hours) and nearly complete closure after 24 hours. Therefore, CGG proved to be a promising material for developing formulations with properties suitable for cutaneous use. F-CGG is more promising as it combines bioadhesion, antioxidant activity, biocompatibility, cell proliferative potential, and enhanced wound healing, indicating that it could be an advanced material for wound treatment.
- What is the chemical structure of Cationic Guar Gum?
Answer: Thank you for your inquiry regarding the chemical structure of Cationic Guar Gum (CGG). Cationic Guar Gum is a modified form of natural guar gum, where the hydroxyl groups in the guar gum backbone are replaced with hydroxypropyltrimonium groups, introducing quaternary ammonium groups into the structure and imparting a positive charge to the polymer. The main structural components comprise alternating mannose and galactose units, typical of guar gum, with attached hydroxypropyltrimonium chloride groups that enhance its solubility, thermal stability, and bioadhesive properties, particularly for cutaneous applications.
In the revised document, we have provided a figure (Figure 1, Page 3) illustrating the CGG chemical structure for further clarification and a comprehensive caption.
- Which UV spectrum is shown in Figure 3E
Answer: Thank you for your comment. The UV spectrum depicted in Figure 3E is the optimized film (6% w/v). To ensure comprehension, we improved the figure’s caption. All modifications are highlighted in green. The current version of the caption is: “Images of films prepared with 1% (A), 2% (B), 4% (C), and 6% (D) (w/v) CGG during development tests and scans obtained in UV-Vis spectrum to F-CGG (6% w/v) (E)” (Lines 424 to 425)
- Figure 5B should illustrate the time for the dry film to be wetted by water. CGG should be hydrophilic in nature, but the wetting time is not enough?
Answer: Thank you for this observation. We agree that additional details on the film's wetting time are valuable for understanding the hydrophilic nature of CGG. While CGG is indeed hydrophilic, our results showed that the wetting behavior depends on the film's specific surface. In Figure 5B, we demonstrated the contact angle of water on the film's upper (air-exposed) and lower (Petri dish-exposed) surfaces, which displayed different behaviors. The upper side showed a more hydrophobic character, likely due to the drying process and interaction with air during film formation. In contrast, the lower side was more hydrophilic, exhibiting a faster wetting time.
Moreover, we would like to clarify that our study did not directly estimate the film's wetting time. Instead, we evaluated the film's swelling behavior, which measures the amount of water the material retains over a specific period. This standard protocol allows us to infer the material's affinity for water and provides a more comprehensive understanding of the film's hydrophilic nature.
The film's swelling index was measured at different time points (6, 12, and 24 hours), and the results demonstrated a high capacity for water absorption, reaching 2032 ± 211% after 6 hours. This behavior confirms the hydrophilic properties of CGG-based films.
We will update the text to clarify the distinction between wetting time and swelling behavior, ensuring the results align with the methodology used in our study. While the contact angle provides insights into surface hydrophilicity, the swelling behavior reflects the material's overall interaction with water more accurately. All modifications were highlighted in green in the manuscript. The following statement was included (Lines 492 to 499): “In this study, we evaluated the swelling behavior of the CGG-based films, which measures the amount of water absorbed and retained by the material over specific time intervals. The result underscores the hydrophilic nature of the CGG-based films. While the contact angle measurements indicated different hydrophilic properties between the upper and lower surfaces of the film, the swelling behavior offers a more robust indication of the material’s interaction with water, particularly relevant for wound healing applications where moisture retention is critical.”
- Figure 6C, is it necessary to keep from water during infrared testing?
Answer: Thank you for your inquiry. Removing water from the sample before FTIR analysis is not necessary. Our equipment, equipped with an ATR (Attenuated Total Reflectance) module, can evaluate samples in various states, including when they are hydrated. This setup allows us to analyze the sample in its original state without dehydration, thereby preserving the sample's natural properties during testing. The ATR-FTIR technique's adaptability in the apparatus makes it well-suited for analyzing materials such as hydrogels and films, even in their hydrated form. Furthermore, before beginning the analysis of the samples, a blank test using either water or an empty device is necessary. This important step removes any potential external interferences from the assessment process.
- The mechanism behind the differences in performance between HC-CGG and F-CCG needs further clarification?
Answer: Thank you for your observation. The differences in performance between the hydrogel (HG-CGG) and film (F-CGG) formulations are primarily due to their compositional variations and preparation steps, which were characterized by several analytical techniques. This allowed us to understand their behavior in wound healing applications better.
HG-CGG is a semisolid formulation with a high-water content, characterized by a porous three-dimensional network. This was assessed using rheological measurements and spreadability tests. These analyses demonstrated the hydrogel’s greater flexibility and ease of application across larger surface areas. However, the high-water content contributes to reduced bioadhesive strength, especially on injured skin, where the increased transepidermal water loss limits the hydrogel's ability to interact effectively with the tissue. This was confirmed by our bioadhesion tests using porcine skin models.
In contrast, F-CGG, a solid form produced through solvent casting, was characterized using mechanical property tests, swelling capacity measurements, and contact angle analysis. These methods highlighted its denser and more compact structure, directly contributing to superior bioadhesive strength, particularly on injured skin. Unlike HG-CGG, the film's structure allows it to maintain contact under conditions of significant water loss. Additionally, F-CGG exhibited higher mechanical strength and a greater swelling index, making it more effective at absorbing wound exudate while maintaining its integrity. The increased concentration of CGG in the film further enhances its antioxidant activity, as more quaternary ammonium groups are available to neutralize free radicals, as demonstrated through the ABTS radical scavenging assay.
While these findings provide valuable insights into the mechanisms behind the differences in performance between HG-CGG and F-CGG, we acknowledge limitations to the possible extrapolations and hypotheses that can be made based on our results. The in vitro nature of our experiments, including bioadhesion, mechanical strength, and antioxidant activity assays, may not fully capture the complexities of in vivo wound healing environments.
However, the evidence gathered from our experiments suggests that the composition differences between HG-CGG and F-CGG play a key role in their distinct performances. We have revised the discussion section of the manuscript to include a more detailed explanation of these mechanisms, while also acknowledging the limitations of our study and the need for further research to confirm these findings in clinical settings.
All modifications were highlighted in green in the manuscript (Lines 619 to 647).
- As the application of wound healing, what are the advantages and disadvantages of CCG compared with other types of hydrogels or membranes?
Answer: Thank you for giving us the opportunity to discuss such aspects of our study. Cationic Guar Gum (CGG)-based film and hydrogel have several advantages compared to conventional wound healing devices and other formulations made of synthetic materials. One key advantage of CGG is its improved bioadhesive properties, which come from the presence of quaternary ammonium groups. These groups interact with the skin, enhancing adhesion and ensuring that the dressing stays in place for longer periods. This is crucial for wound healing, as maintaining a moist environment at the wound site is essential for faster tissue regeneration. In contrast, many synthetic hydrogels or conventional dressings often lack sufficient adhesion to the skin, especially when the wound is exuding fluids, which may lead to frequent dressing changes and discomfort for the patient (10.3390/pharmaceutics15030874).
Furthermore, CGG is highly hydrophilic, allowing it to absorb large amounts of wound exudate while retaining moisture, which is vital for creating an ideal wound-healing environment. In comparison, synthetic hydrogels, while effective in moisture retention, sometimes have limited swelling capacities and may not hold as much fluid, particularly in heavily exudative wounds. CGG is also naturally biocompatible and biodegradable and exhibits no cytotoxicity, reducing the risk of adverse reactions during wound healing. Many synthetic materials, although functional, may not offer the same level of biodegradability or biocompatibility, potentially leading to the need for more invasive removal processes or causing irritation in sensitive patients.
Another advantage of CGG-based formulations is their antioxidant activity, which has been shown to neutralize free radicals at the wound site, promoting a healthier healing environment. Synthetic hydrogels, while useful in maintaining moisture and protecting wounds, generally do not offer intrinsic antioxidant properties, making CGG a more multifunctional option. However, CGG does have certain disadvantages compared to synthetic alternatives. For instance, its mechanical strength is relatively lower, making it less suitable for wounds in areas of the body that experience frequent movement or mechanical stress. Synthetic hydrogels, or even conventional wound dressings such as silicone-based products, often have superior tensile strength and durability, making them more appropriate for certain injuries.
Nevertheless, CGG's biocompatibility, biodegradability, and multifunctionality make it an attractive option for wound healing, especially compared to conventional devices and synthetic hydrogels, which may not offer the same level of bioactivity or comfort for the patient.
To better explore such aspects, we included a brief state concerning the advantages and disadvantages of CGG-based formulations prepared in this study in comparison to conventional wound devices. All modifications were highlighted in green in the manuscript. The following statement was included (Lines 707 to 724): “When considering the use of CGG for wound healing, it is important to highlight its advantages and limitations compared to traditional wound healing devices and synthetic hydrogels. CGG is notable for its strong bioadhesive properties due to the presence of quaternary ammonium groups. These properties ensure prolonged contact with the wound site, creating a moist environment that is critical for tissue regeneration. Additionally, CGG-based hydrogels and films demonstrate exceptional hydrophilicity, with a high swelling capacity that allows them to absorb large volumes of wound exudate while maintaining moisture. This sets CGG apart from synthetic hydrogels, which may not achieve the same level of fluid retention or bioactivity. Furthermore, CGG is a naturally biocompatible and biodegradable material, ensuring a safer interaction with biological tissues without the cytotoxicity or removal difficulties associated with synthetic alternatives. However, CGG has lower mechanical strength, making it less suitable for wounds in areas under frequent movement or stress, where synthetic hydrogels or silicone-based dressings might be more durable. Additionally, CGG hydrogels have shown reduced bioadhesion on injured skin, indicating the need for further formulation optimization. Nonetheless, CGG's multifunctionality, including its antioxidant activity, positions it as a promising material for wound healing applications, offering unique benefits compared to conventional and synthetic alternatives.”

Reviewer 2 Report
Comments and Suggestions for Authors
In the following manuscript Kamila Gabrieli Dallabrida, et al. are regarding about cationic guar gum as innovative hydrogels and films. The materials are proposed as to use in wound healing.
It is research article composed of Abstract, Introduction, Materials and Methods, Results and Discussion, and Conclusions.
The introduction is extensive and closely related to the topic. The authors cited 73 references and each of them (except one) is from last 20 years, what makes the manuscript current and closely related to current trends in wound healing.
Generally the manuscript is well organized, the experiment is very interesting, and the research is very advanced, But I have some comments about the article.
1. Why are the authors studying these materials in both hydrogel and film forms?
2. The authors did not use any cross-linking agents in the preparation of the materials, and the gum they used is water-soluble. This raises concerns about the stability of these materials in aqueous conditions, particularly over extended periods. Specifically, I am curious about the stability beyond the 24-hour swelling test period reported by the authors. At what point does material degradation, especially of the films, begin to occur under exposure to water?
3. Why are the authors studying these materials in the form of a hydrogel? Typically, hydrogels used as wound dressings are more like porous patches or pieces of polymeric material in a single, solid form that can be applied directly to the wound. In this case, the hydrogel is liquid. How do they envision its use in the future?
4. Was the physical stability tested on the same samples that were used for the pH measurements?
5. Please describe in more detail the purpose of the centrifugation test.
6. Line 337, the authors mention the resistance properties of the hydrogel. Which test supports this claim? How can the resistance properties of the hydrogel be assessed in the form in which the authors obtained hydrogel?
7. Details about chemicals, procedures and equipment should be presented (e.g. company and country of company). Please read the article carefully.
Thank you
Author Response
Reviwer 2
In the following manuscript Kamila Gabrieli Dallabrida, et al. are regarding about cationic guar gum as innovative hydrogels and films. The materials are proposed as to use in wound healing. It is research article composed of Abstract, Introduction, Materials and Methods, Results and Discussion, and Conclusions.
The introduction is extensive and closely related to the topic. The authors cited 73 references and each of them (except one) is from last 20 years, what makes the manuscript current and closely related to current trends in wound healing. Generally the manuscript is well organized, the experiment is very interesting, and the research is very advanced, But I have some comments about the article.
Answer: We are grateful for the reviewer's positive feedback on the structure, relevance, and progression of our research. We are delighted to learn that the thoroughness of the introduction and its alignment with current trends in wound healing has been acknowledged. The selection of recent references was purposeful, as we aimed to align the manuscript with the latest developments and emerging concepts in the field. We have carefully reviewed the comments provided and have addressed each one individually, making the necessary revisions to enhance the clarity and quality of the manuscript. Thank you once again for your thoughtful review. All modifications were highlighted in green in the manuscript.
- Why are the authors studying these materials in both hydrogel and film forms?
Answer: Thank you for your question. We chose to explore Cationic Guar Gum (CGG) in both hydrogel and film forms due to their distinct advantages in wound healing applications. Hydrogels, with their high water content, offer a cooling and soothing effect, reduce pain upon application, and maintain a moist environment that promotes wound healing. Their porous and flexible structure makes them well-suited for covering large or irregular wound surfaces, especially for wounds requiring frequent dressing changes.
Conversely, films provide greater mechanical strength and longer adhesion to the wound site, making them suitable for wounds where reduced dressing changes are preferred. Films also offer better control over the release of therapeutic agents and act as a protective barrier against external contaminants, while their transparency allows continuous monitoring of the wound without removing the dressing. By assessing CGG in both forms, we aimed to explore its versatility and determine which formulation is more effective for specific clinical needs.
Our findings indicate that the film form of CGG demonstrates superior performance in terms of bioadhesion, mechanical strength, and antioxidant activity. This suggests that CGG may be better suited for the preparation of solid, long-lasting wound care materials. Additionally, our study is the first to investigate CGG in both hydrogel and film formats, offering valuable insights into its potential applications. We believe that this research will be a valuable resource for future studies, showcasing the diverse applications of natural gums in the development of therapeutic and wound healing formulations.
All modifications were highlighted in green in the manuscript (Lines 746 to 758).
- The authors did not use any cross-linking agents in the preparation of the materials, and the gum they used is water-soluble. This raises concerns about the stability of these materials in aqueous conditions, particularly over extended periods. Specifically, I am curious about the stability beyond the 24-hour swelling test period reported by the authors. At what point does material degradation, especially of the films, begin to occur under exposure to water?
Answer: Thank you for highlighting this important point. It is indeed accurate that we did not utilize cross-linking agents in the preparation of the Cationic Guar Gum (CGG) formulations, and it is confirmed that CGG is water-soluble. The stability of these materials in aqueous environments is a critical factor for their potential application in wound healing. Our study specifically examined the swelling behavior of the film over a 24-hour period, demonstrating its capacity to absorb water and maintain its structure during that time. However, we did not extensively test the long-term stability of the film beyond this initial period.
Based on the properties of CGG and our observations from the swelling test, we anticipate that, without cross-linking, the material may begin to exhibit signs of degradation after prolonged exposure to water, likely beyond the 24-hour period. The absence of cross-linking agents suggests that the films could gradually lose their mechanical integrity as the polymer chains interact with and absorb water over extended periods, potentially resulting in softening or dissolution.
This study is preliminary, and we acknowledge the need to further investigate various aspects of the material's long-term stability, including its behavior under prolonged aqueous exposure. Additionally, exploring the microbiological stability of the films is crucial, especially for clinical applications in wound healing. To address this, future studies should consider including preservatives in the film composition to enhance its resistance to microbial contamination over time.
Future studies should involve extended stability testing under aqueous conditions, assessing both the mechanical and structural properties over time to comprehensively evaluate the films' long-term stability and potential degradation. We recognize that incorporating cross-linking agents or other stabilizing strategies could enhance the durability of CGG-based materials for applications requiring prolonged exposure to moisture.
We have revised the discussion section of the manuscript to reflect these limitations and highlight potential modifications that could improve the mechanical and microbiological stability of CGG-based formulations in future applications.
All modifications were highlighted in green in the manuscript (Lines 727 to 736).
- Why are the authors studying these materials in the form of a hydrogel? Typically, hydrogels used as wound dressings are more like porous patches or pieces of polymeric material in a single, solid form that can be applied directly to the wound. In this case, the hydrogel is liquid. How do they envision its use in the future?
Answer: Thank you for your question. It is important to note that we did not prepare a liquid hydrogel in our study. Before formulating the hydrogel, we conducted a pre-formulation stage to determine the optimal concentration of Cationic Guar Gum (CGG) for the desired spreadability and viscosity for topical application. This meticulous process ensured that the hydrogel would be easy to apply to the skin, striking the right balance between being too fluid or too viscous, thus enhancing its effectiveness in wound healing applications. Our focus on pre-formulation ensures that the hydrogel's physical properties are compatible with various applications, providing a strong foundation for future development. The resulting semisolid hydrogel formulation possesses characteristics akin to common cutaneous products, providing ample coverage and moisture retention at the wound site as well as high bioadhesiveness.
As we move forward with this preliminary study to demonstrate the use of CGG in the production of wound healing devices, we will also explore further optimizations to the formulation. In future investigations, our goal is to integrate active substances to potentiate the per se wound healing potential identified to CGG and substances that can serve dual functions, such as acting as plasticizers to films while also offering pharmacological effects. For example, incorporating aloe vera, as demonstrated in the study by Alves and collaborators (2024) (10.1016/j.jddst.2024.105982), could potentially enhance the wound-healing properties of the formulation. This strategic approach will enable us to maximize the therapeutic potential of the hydrogel while upholding the desired mechanical and bioadhesive properties for effective wound care.
For enhancing comprehension, we included a brief statement in the revised manuscript (Lines 736 to 745).
References
Alves, F.M.S.; El Zein, A.K.; Cobre, A. de F.; Lazo, R.E.L.; Reolon, J.B.; Marchiori, C.; Costa, J.S. da; Pontarolo, R.; Fajardo, A.R.; Sari, M.H.M.; et al. Aloe Vera Miller Extract as a Plasticizer Agent to Polymeric Films: A Structural and Functional Component. J Drug Deliv Sci Technol 2024, 99, 105982, doi:10.1016/J.JDDST.2024.105982.
- Was the physical stability tested on the same samples that were used for the pH measurements?
Answer: Thank you for your question. In our study, all tests, including physical stability and pH measurements, were carried out in triplicate on three different batches of the formulations. This approach enabled us to capture any potential variability between batches while maintaining high consistency across all evaluations. Despite using different batches, all tests were performed on every formulation batch to ensure a comprehensive assessment of both physical stability and pH behavior. This method allowed us to verify the reproducibility of our results across independent preparations. It ensured that any observed changes in stability or pH were indicative of the formulation’s overall behavior, rather than being influenced by a single batch.
For enhancing comprehension, we included a brief statement regarding the conditions of our assays and batch preparation (Line 175 to 176).
- Please describe in more detail the purpose of the centrifugation test.
Answer: Thank you for giving us the opportunity to improve such an aspect. The centrifugation test was conducted to evaluate the formulations' physical stability, specifically their ability to resist phase separation under accelerated conditions. Centrifugation applies a force much greater than that of gravity, simulating the effects of long-term storage in a shorter time frame. This test is commonly used to assess the potential for separation of the components in semi-solid formulations, such as hydrogels, where issues like sedimentation or creaming might occur over time.
By subjecting the formulations to centrifugation, we aimed to observe whether the samples could maintain their homogeneity under stress, which would serve as an indicator of their stability during storage and handling. If a formulation exhibits phase separation during centrifugation, it may be prone to instability under normal storage conditions. The absence of phase separation in our formulations after centrifugation suggests that they possess adequate physical stability, making them suitable for wound healing applications where consistent performance is critical.
For enhancing comprehension, we included a brief statement in the revised manuscript (Lines 155 to 161).
- Line 337, the authors mention the resistance properties of the hydrogel. Which test supports this claim? How can the resistance properties of the hydrogel be assessed in the form in which the authors obtained hydrogel?
Answer: Thank you for your input. We acknowledge that the wording in the manuscript was incomplete and unclear. The reference to "resistance properties" was intended to describe a preliminary evaluation conducted during the pre-formulation stage, where we empirically assessed the concentration of Cationic Guar Gum (CGG). Our aim at this stage was to identify an optimized formulation with the appropriate balance between fluidity and viscosity. We visually assessed the formulations to ensure they did not exhibit excessive fluidity or overly high viscosity, which could hinder their applicability in wound healing. After identifying the concentration that provided the best balance of properties, we proceeded with the formal testing of the selected formulation. We have revised the manuscript to clarify this point and ensure that the process is accurately reflected (Lines 351 to 356).
- Details about chemicals, procedures and equipment should be presented (e.g. company and country of company). Please read the article carefully.
Answer: Thank you for your feedback. We acknowledge the importance of providing comprehensive details about the chemicals, procedures, and equipment used in our study, including the suppliers and their countries of origin. After a thorough review of the manuscript, we have made the necessary updates to ensure that all relevant materials and equipment are now included in the methods section (highlighted in green). This enhancement will improve the reproducibility of the study and provide greater transparency regarding the materials' sources.

Round 2
Reviewer 1 Report
Comments and Suggestions for Authors
No.
Reviewer 2 Report
Comments and Suggestions for Authors
The authors have responded satisfactorily to my comments and remarks. In its present form I recommend this article for publication.